# Disease characteristics and management of hospitalised adolescents and adults with community-acquired pneumonia in China: a retrospective multicentre survey

Liang Chen,[1,2] Fei Zhou,[1] Hui Li,[3] Xiqian Xing,[4] Xiudi Han,[5] Yiming Wang,[3] Chunxiao Zhang,[6] Lijun Suo,[7] Jingxiang Wang,[8] Guohua Yu,[9] Guangqiang Wang,[10] Xuexin Yao,[11] Hongxia Yu,[12] Lei Wang,[13] Meng Liu,[1] Chunxue Xue,[1] Bo Liu,[7] Xiaoli Zhu,[14] Yanli Li,[1] Ying Xiao,[1] Xiaojing Cui,[3] Lijuan Li,[3] Timothy M Uyeki,[15] Chen Wang,[3,16] Bin Cao,[3,16] CAP-China network

For numbered affiliations see end of article.

**Correspondence to**
Dr Bin Cao;
Caobin_ben@163.com

## ABSTRACT

**Objectives** To describe the clinical characteristics and management of patients hospitalised with community-acquired pneumonia (CAP) in China.

**Design** This was a multicentre, retrospective, observational study.

**Setting** 13 teaching hospitals in northern, central and southern China from 1 January 2014 to 31 December 2014

**Participants** Information on hospitalised patients aged ≥14 years with radiographically confirmed pneumonia with illness onset in the community was collected using standard case report forms.

**Primary and secondary outcome measures** Resource use for CAP management.

**Results** Of 14 793 patients screened, 5828 with radiographically confirmed CAP were included in the final analysis. Low mortality risk patients with a CURB-65 score 0–1 and Pneumonia Severity Index risk class I–II accounted for 81.2% (4434/5594) and 56.4% (2034/3609) patients, respectively. 21.7% (1111/5130) patients had already achieved clinical stability on admission. A definite or probable pathogen was identified only in 12.7% (738/5828) patients. 40.9% (1575/3852) patients without pseudomonal infection risk factors received antimicrobial overtreatment regimens. The median duration between clinical stability to discharge was 5.0 days with 30-day mortality of 4.2%.

**Conclusions** These data demonstrated the overuse of health resources in CAP management, indicating that there is potential for improvement and substantial savings to healthcare systems in China.

**Trial registration number** NCT02489578; Results.

## BACKGROUND

Community-acquired pneumonia (CAP) is one of the most common infectious syndromes and is a leading cause of death worldwide.[1 2] In Europe, the reported

### Strengths and limitations of this study

► This is the largest multicentre study to investigate demographic characteristics, severity and microbiological testing, empirical antimicrobial treatment, duration of hospitalisation and 30-day mortality among adults and adolescents hospitalised with community-acquired pneumonia (CAP) in mainland China, including adolescents and adults of all ages admitted to general hospital wards or intensive care units from the participating centres, patients who were critically ill and aged >90 years.

► The participating hospital sites are teaching hospitals in seven cities in three provinces, and may not be representative of CAP in smaller, rural hospitals.

► The majority of patients are adult patients with CAP, so our findings do not apply to children hospitalised with CAP.

rate of CAP ranges from 1.6 to 9 cases per 1000 in the general adult population per year.[3–5] Despite advances in medical technology and global economic development, CAP-associated mortality remains high (eg, 20.9/100 000 in the USA and 12.7/100 000 in Canada).[1 2] Patients hospitalised in intensive care units (ICUs) for CAP have mortality in excess of 20% for immunocompetent patients and closer to 30% for those immunocompromised.[6] In Japan and Korea, the 30-day mortality of patients hospitalised with CAP is about 4%–6%.[7 8]

Although mainland China has nearly 19% of the world's population, there are limited data on CAP management and disease burden in China during the last 10 years. According

to a household interview survey published in the China Health and Family Planning Statistical Yearbook (2013), the 2-week prevalence of pneumonia in China was estimated to be 11/1000, and the direct cost due to bacterial pneumonia was about 320 million RMB (approximately $46.4 million).[9] In 2015, CAP-China, a multicentre clinical network, was founded with the support of National Key Technology Support Program from Ministry of Science and Technology (2015BAI12B11) to provide data on CAP for clinical researchers and healthcare policy makers in China.

A multicentre retrospective study of all hospitalised patients with CAP from 13 centres in northern, central and southern China among CAP-China members was implemented in 2015 (clinical trial registration no. NCT02489578, Results). To our knowledge, this is the largest multicentre study to investigate demographic characteristics, severity and microbiological testing, empirical antimicrobial treatment, duration of hospitalisation and 30-day mortality among adults and adolescents hospitalised with CAP in mainland China.

## METHODS

### Study design and population

Data were collected from 13 hospitals in Northern (Beijing), Central (Yantai, Qindao, Weifang, Zibo, Rizhao cities in Shandong Province) and Southern (Kunming City in Yunan Province) China. A listing of participating centres can be found in online supplementary appendix 1. All patients admitted to the 13 hospitals during 1 January 2014 through 31 December 2014 with the relevant disease codes of pneumonia or pulmonary infection in the WHO International Classification of Diseases 10th revision (online supplementary appendix 2) were eligible. Data on all eligible patients identified in screening were retrieved from the hospital information system in each centre. Trained physicians reviewed the medical case history and collected data on 786 variables for each patient. Chest radiographs and CT scans for each patient were reviewed by pulmonary physicians and radiologists in each centre. Two-level review process was performed for data collection and entry.

The CAP case definition includes (1) illness onset in the community (defined as community-acquired infection among those who have not been hospitalised during recent 28 days)[10]; (2) chest radiograph or CT scan showing infiltrate or interstitial changes, with or without pleural effusion; (3) any one of pneumonia clinical manifestations: (1) recent cough, sputum or aggravation of respiratory symptoms, the emergence of purulent sputum, with or without chest pain; (2) fever (defined as axillary temperature≥37.3°C)[11] or hypothermia (axillary temperature<36°C); (3) signs of pulmonary consolidation and (or) moist crackles; or (4) white cell count >10×10$^9$/L or <4×10$^9$/L, with or without neutrophil predominance.

Patients were excluded if (1) age <14 years; (2) pneumonia onset ≥48 hours after admission; (3) lung infiltrate or interstitial changes that were interpreted as lung cancer, pulmonary tuberculosis, non-infectious interstitial lung diseases, pulmonary oedema, atelectasis, pulmonary embolism, pulmonary eosinophil infiltrate and pulmonary vasculitis; (4) immunocompromised status (including HIV(+), chemotherapy/radiotherapy within 6 months, immunosuppressive therapy, organ/bone marrow transplantation, splenectomy, haematological neoplasms); (5) readmission within 72 hours after discharge.

### Quality control of the study

Key investigators, including clinicians, statisticians, microbiologists and radiologists, worked together to draft the protocol and created a single formatted case report form (CRF) that was used by all centres. Before study initiation, all investigators from the 13 centres received training on the protocol, screening process, definition of underlying diseases and formatted CRF (online supplementary appendix 3). After data were collected, the CRF was reviewed by a trained researcher to ensure its completeness and data quality. A second review was performed independently by a trained team of physicians in each centre before being entering in duplicate into a computerised database.

### Data collection

A total of 786 variables were included in the formatted CRF, including:

1. demographic data: age, gender, ID number, source of admission, types of medical insurance;
2. underlying diseases: chronic lung, heart, renal and liver diseases, diabetes, hypertension, solid organ cancers. Definition of underlying diseases is listed in online supplementary appendix file 4;
3. factors for acquisition or prevention of CAP: pregnancy, post partum within 6 months, current smoking history, excessive drinking, exposure to day care centre children, bed-ridden longer than 2 months, chronic receipt of corticosteroids (dosage equivalent prednisolone ≥10 mg/day for more than 30 days), statin use, *Streptococcus pneumonia* or influenza vaccination within 1 year;
4. clinical manifestations, clinical signs: recorded on the day of admission, on the fourth hospital day, change of antibiotics within 14 days of admission and the day of discharge or death. Laboratory and radiological findings were also recorded if such tests were repeated by attending physicians. Pneumonia disease severity scores (Pneumonia Severity Index (PSI)/CURB-65) were also recorded;
5. microbiological examination: Gram stain and culture of sputum within 48 hours, blood culture within 48 hours, Bronchoalveolar lavage (BALF) and pleural fluid culture within 1 week after admission, serum antibody (including IgM and IgG) for atypical pathogens (*Mycoplasma pneumoniae, Chlamydia pneumoniae, Legionella pneumophila*). Urinary antigen

testing was performed for *S pneumonia* and *Legionella* spp. Real-time PCR testing was done for respiratory virus and atypical pathogens with sputum and BALF. Nasopharyngeal swab was used for antigen testing for influenza A and influenza B. Aspirate was not routinely used for antigen testing;

6. antimicrobial treatment before admission and change of antimicrobials during hospitalisation. Use of corticosteroids, vasopressors, mechanical ventilation, continuous renal replacement therapy and extracorporeal membrane oxygenation were also recorded;

7. clinical stability was defined as satisfying all of the following: axillary temperature ≤37.8°C for more than 24 hours without use of antipyretic medications; resting heart rate ≤100 beats/min; respiratory rate ≤24 breaths/min; systolic blood pressure ≥90 mm Hg; $SpO_2$ ≥90% on room air; ability to maintain oral intake; normal mental status[12];

8. overtreatment was defined as (1) use of antipseudomonal β-lactams or β-lactams+fluoroquinolones in hospitalised (not in ICU) patients without risk factors for pseudomonal infection; (2) use of β-lactams (antipseudomonal or not)+fluoroquinolones in ICU patients aged <65 years without risk factors for *Pseudomonas* infection; (3) use of anti-Methicillin-resistant Staphylococcus aureus (MRSA) drugs in hospitalised (not in ICU) patients (use of anti-MRSA drugs in ICU patients with MRSA risk after influenza virus infection was considered adequate)[13];

9. risk factors for pseudomonal infection was defined as chronic airway disease (bronchiectasis and chronic obstructive pulmonary disease (COPD)) and at least one risk factor for healthcare-associated pneumonia (HCAP) as defined by the 2005 Infectious Diseases Society America/American Thoracic Society (IDSA/ATS) adult CAP guidelines[13–17];

10. empirical antimicrobial regimens recommended by Chinese CAP guidelines were showed in online supplementary appendix 5.

## Microbiology testing

The conditions that a pathogen was defined as the definite or probable aetiology based on were showed in online supplementary appendix 6.

## Statistical analysis

No formal sample size calculations were performed because of the retrospective descriptive study design. All data were analysed by descriptive statistics with SPSS V.19. Measurement data were tested for normality by Kolmogorov-Smirnov. Measurement data of normal distribution was reported as mean±SD. Measurement data of non-normal distribution was reported as median. The $\chi^2$ test statistics were used for 30-day mortality subgroup analysis. A P value of <0.05 was considered statistically significant.

## RESULTS

### Screening process

A total of 14 793 patients were screened to meet the inclusion and exclusion criteria for CAP, and 5828 patients were included in the final analysis (online supplementary appendix figure 1).

### Epidemiological characteristics

The proportions of male and female patients were similar. The median age was 65 years, range 14–103 years. Prevalent comorbidities included hypertension (35.2%), coronary heart disease (20.0%), diabetes (15.7%), cerebrovascular diseases (15.3%) and COPD (13.7%). 14.9% of patients with CAP had at least one HCAP risk factor (according to IDSA/ATS hospital-acquired prenumonia (HAP)/HCAP guideline published in 2005[14]). 45.7% of patients received antibiotics before admission.

A substantial proportion of admitted patients had relatively mild disease as indicated by the following: (1) CURB-65 score[18] 0–1 accounted for 81.2%, (2) PSI risk class[19] I–II accounted for 56.3%; (3) Shorr Score[20] 0–1 accounts for 99.6%; and (4) Aliberti Score[21] low risk group in 89.7%; (5) only 12.0% (261/2172) patients had procalcitonin more than 2 ng/mL; (6) as many as 65.7% (3741/5698) patients had normal peripheral leucocyte counts (4000–10 000/μL). Most importantly, 21.7% of patients had met criteria for clinical stability at hospital admission[12] (tables 1 and 2).

### Clinical and radiological features

Clinical and radiological features on admission are shown in table 2. Cough, sputum, shortness of breath and fever were the most common. 64.8% of patients had multilobar infiltrates and 20.7% of patients had pleural effusion.

### Microbiological testing

75.0% of patients had some types of microbiological testing. 68.9% of patients had a sputum culture obtained within 48 hours of admission, although only 18.5% of patients were able to produce a sputum culture of acceptable quality. The proportion of patients with blood culture, BALF culture and pleural effusion culture were 10.3%, 9.1% and 1.9%, respectively. Only 0.8% of patients had a urinary antigen test sent to evaluate for *L pneumophila*, and 2.6% had urinary antigen testing for *S pneumoniae* (table 3).

Of all patients, serological testing for antibodies to *M pneumoniae* was only performed on a single serum specimen for IgM (31.2%) and IgG antibodies (13.6%). Similarly, serological testing on a single serum specimen was done for *C pneumoniae* IgM antibody in 22.2% of patients and for IgM antibodies to *L pneumophila* and respiratory viruses in 11.1%. No convalescent serum specimens were collected for serological testing for any pathogens, limiting interpretation of serology results for a single serum specimen.

A definite or probable pathogen was identified only in 12.7% of patients (738/5828): only bacteria

**Table 1** Demographic characteristics and underlying diseases

| Items | Cases (%) |
|---|---|
| Male | 3117 (53.5) |
| Age (years, median, IQR) | 65 (53–78) |
| 14–64 | 2802 (48.1) |
| 65–74 | 1081 (18.5) |
| 75–89 | 1760 (30.2) |
| ≥90 | 185 (3.2) |
| Source of admission (n=5823) | |
| From outpatient department | 4183 (71.8) |
| From emergency room | 1588 (27.3) |
| Transfer from other hospital | 52 (0.9) |
| Days from illness onset to admission (n=5826, median, IQR) | 6.0 (3.0–14.0) |
| Patients who received antibiotics before admission | 2664 (45.7) |
| β-Lactams | 1015 (38.1) |
| Fluoroquinolones | 586 (22.0) |
| Macrolides | 170 (6.4) |
| β-Lactams+fluoroquinolones | 413 (15.5) |
| β-Lactams+macrolides | 201 (7.5) |
| Others | 279 (10.5) |
| Systemic glucocorticosteroids use before admission | 250 (4.3) |
| Underlying diseases | 4219 (72.4) |
| Hypertension | 2053 (35.2) |
| Coronary heart disease | 1163 (20.0) |
| Diabetes | 913 (15.7) |
| Cerebrovascular diseases | 890 (15.3) |
| COPD | 801 (13.7) |
| Bronchiectasis | 629 (10.8) |
| Asthma | 339 (5.8) |
| Malignant solid tumours | 254 (4.4) |
| Congestive heart failure | 202 (3.5) |
| Chronic renal diseases | 201 (3.4) |
| Connective tissue diseases | 110 (1.9) |
| Chronic hepatic diseases | 90 (1.5) |
| Smoking status | |
| Current smokers | 1009 (17.3) |
| Ex-smokers | 590 (10.1) |
| Alcoholism | 407 (7.0) |
| Risk factors for aspiration* | 377 (6.5) |
| History of CAP within 1 year | 368 (6.3) |
| History of vaccination | |
| Influenza vaccine within 1 year | 12 (0.2) |
| *Streptococcus pneumoniae* vaccine within 5 years | 8 (0.1) |
| Risk factors for HCAP according to IDSA/ATS criteria | 868 (14.9) |
| Hospitalised in an acute care hospital for two or more days within 90 days | 404 (6.9) |
| Received recent intravenous antibiotic therapy, chemotherapy or wound care within the past 30 days | 656 (11.3) |
| Attended a hospital or haemodialysis clinic | 36 (0.6) |
| Residence in a nursing home or long-term care facility | 19 (0.3) |

Continued

**Table 1** Continued

| Items | Cases (%) |
|---|---|
| CURB-65 score (n=5594) | |
| 0 | 2343 (41.9) |
| 1 | 2199 (39.3) |
| 2 | 884 (15.8) |
| 3 | 147 (2.6) |
| 4 | 20 (0.4) |
| 5 | 1 (0.0) |
| PSI risk class (n=3609) | |
| I | 1130 (31.3) |
| II | 904 (25.0) |
| III | 748 (20.7) |
| IV | 646 (17.9) |
| V | 181 (5.0) |
| Shorr Score (n=5650) | |
| 0 | 5084 (90.0) |
| 1 | 541 (9.6) |
| 2 | 23 (0.4) |
| 3 | 2 (0.0) |
| 4 | 0 (0.0) |
| Aliberti Score (n=5828) | |
| Low-risk group | 5226 (89.7) |
| High-risk group | 602 (10.3) |
| Clinical stability on admission† (n=5130) | 1111 (21.7) |

*Risk factors for aspiration included choking, drowning, nasal feeding, pseudobulbar palsy, dementia, coma, poisoning and Parkinson's disease.
†Clinical stability was defined as satisfying the following at the same time: axillary temperature ≤37.8°C for more than 24 hours; heart rate ≤100 beats/min in resting state; breathing rate ≤24 breaths/min; systolic blood pressure ≥90 mm Hg; $SpO_2$ ≥90% on room air; ability to maintain oral intake; normal mental status.
CAP, community-acquired pneumonia; COPD, chronic obstructive pulmonary disease; CURB-65, confusion; urea >7 mmol/L, respiratory rate >30 breaths/min, systolic blood pressure <90 mmHg or diastolic blood pressure <60 mmHg, age ≥65 years; HCAP, healthcare associated pneumonia; IDSA/ATS, Infectious Diseases Society America/American Thoracic Society; PSI, Pneumonia Severity Index.

in 87.1% (643/738), only atypical pathogens in 0.9% (7/738), only viruses in 8.5% (63/738), bacteria and viruses in 2.7% (20/738), viruses and atypical pathogens in 0.7% (5/738). The most common five pathogens identified were *Pseudomonas aeruginosa* 26.7% (197/738), *Klebsiella pneumonia* 17.6% (130/738), *Escherichia coli* 8.9% (66/738), *Acinetobacter* 8.4% (62/738) and influenza A virus 7.3% (54/738) (online supplementary appendix 7).

### Empiric antimicrobial regimens

β-Lactams (received by 72.7% of patients) and fluoroquinolones (received by 42.2%) were the most common classes of antibiotics that were administered empirically. In patients (not in ICU) without pseudomonal infection risk factors, 27.8% (1070/3852) of patients received empiric antibiotic regimens including antipseudomonal β-lactams, and 12.1% (468/3852) of patients received β-lactams+fluoroquinolones; 0.4% (16/3852) of patients

**Table 2** Clinical and radiological features on admission

| Items | Cases (%) |
|---|---|
| Axillary temperature ≥38°C (n=5826) | 2783 (47.8) |
| Axillary temperature <36°C(n=5793) | 44 (0.8) |
| Cough | 5192 (89.1) |
| Sputum | 4751 (81.5) |
| Shortness of breath | 2116 (36.3) |
| Chest pain | 709 (12.2) |
| Decrease of consciousness | 294 (5.0) |
| Chest signs | |
| Moist rales | 2919 (50.1) |
| Dry rales | 1387 (23.8) |
| Oedema of lower limbs | 592 (10.2) |
| Cyanosis | 547 (9.4) |
| SBP <90 mm Hg | 45 (0.8) |
| Radiology | |
| Infiltrate more than two lobes | 3776 (64.8) |
| Plural effusion | 1205 (20.7) |
| Cavitation | 228 (3.9) |
| WBC (×10^9/L , n=5698) | |
| >10.0 | 1626 (28.5) |
| <4.0 | 331 (5.8) |
| 4.0–10.0 | 3741 (65.7) |
| BUN >7.0 mmol/L (n=5601) | 1166 (20.8) |
| pH <7.30 (n=3330) | 87 (2.6) |
| PaO$_2$/FiO$_2$ <300 mm Hg (n=3327) | 1196 (35.9) |
| PCT (ng/mL, n=2172) | |
| PCT ≤0.25 | 1307 (60.2) |
| 0.25<PCT<1 | 479 (22.1) |
| 1≤PCT<2 | 125 (5.8) |
| PCT ≥2 | 261 (12.0) |

BUN, blood urea nitrogen; PaO$_2$/FiO$_2$, arterial pressure of oxygen/fraction of inspiration oxygen; PCT, procalcitonin; pH, potential of hydrogen; SBP, systolic blood pressure; Scr, serum creatinine; WBC, white blood cell.

**Table 3** Microbiological examination for CAP

| Items | Cases (%) |
|---|---|
| Any microbiological examination | 4371 (75.0) |
| Microbiological examination for bacteria | 4015 (68.9) |
| Microbiological examination for atypical aetiology | 1983 (34.0) |
| Microbiological examination for virus | 2014 (34.6) |
| Bacterial or fungal culture | 4015 (68.9) |
| Qualified sputum culture* | 1078 (18.5) |
| Blood culture† | 602 (10.3) |
| BALF culture*‡ | 532 (9.1) |
| Pleural effusion culture† | 108 (1.9) |
| Antibody-based assays on acute serum | |
| *Mycoplasma pneumoniae* | IgM: 1821 (31.2) IgG: 794 (13.6) |
| *Chlamydia pneumoniae* | IgM: 1294 (22.2) IgG: 220 (3.8) |
| *Legionella pneumoniae* | IgM: 645 (11.1) IgG: 227 (3.9) |
| Adenovirus | IgM: 644 (11.1) IgG: 0 (0.0) |
| Respiratory syncytial virus | IgM: 643 (11.0) IgG: 0 (0.0) |
| Influenza A virus | IgM: 643 (11.0) IgG: 0 (0.0) |
| Influenza B virus | IgM: 640 (11.0) IgG: 0 (0.0) |
| Parainfluenza virus | IgM: 643 (11.0) IgG: 0 (0.0) |
| Nucleic acid-based molecular diagnostics | |
| From sputum | 297 (5.1) |
| Time interval§ (days, median, IQR) | 9.0 (6.0–16.0) |
| From BALF‡ | 19 (0.3) |
| Time interval§ (days, median, IQR) | 13.0 (9.0–24.0) |
| *M. pneumoniae* | 270 (4.6) |
| *Chlamydia* spp | 270 (4.6) |
| *Legionella* spp | 270 (4.6) |
| Influenza A virus | 270 (4.6) |
| Influenza B virus | 270 (4.6) |
| Other respiratory virus¶ | 270 (4.6) |
| Urinary antigen test | |
| *S pneumoniae* | 150 (2.6) |
| *Legionella* spp | 47 (0.8) |
| Nasopharyngeal swab antigen testing | |
| Influenza A virus | 41 (0.7) |
| Influenza B virus | 21 (0.4) |

*Within 48 hours after admission.
†Within 1 week after admission.
‡BALF, bronchoalveolar lavage fluid.
§Days from illness onset to testing.
¶Parainfluenza virus (PIV) types 1, 2, 3 and 4, rhinovirus (HRV), enterovirus (EV), coronovirus (hCoV) types 229E, NL63, OC43 and HKU1, parapneumovirus (hMPV), and adenovirus (AdV), bocavirus.
CAP, community-acquired pneumonia.

aged <65 years and not in ICU received β-lactams (antipseudomonal or not)+fluoroquinolones combined regimens. Overall, 40.9% (1575/3852) of patients without pseudomonal infection risk factors received antimicrobial overtreatment regimens (table 4).

## Clinical outcomes

Clinical outcomes are shown in table 5. Overall, 6.3% of patients were admitted to an ICU, and 2.7% required invasive mechanical ventilation. Vasopressors were administered to 3.4% of patients, and 26.4% received corticosteroids during the hospitalisation. The 30-day mortality was 4.2%. The median duration of hospitalisation was 11 days. The median duration from admission to clinical stability was 4 days, and from clinical stability to

**Table 4** Empirical antimicrobial regimen for patients with CAP (n=5716)*

| Empirical antimicrobials (%) | Without risk factors for *Pseudomonas* infection (n=3852) | | | | With risk factors for *Pseudomonas* infection (n=1864)† |
|---|---|---|---|---|---|
| | Age <65 years and not in ICU (n=1881) | Age <65 years and in ICU (n=79) | Age ≥65 years and not in ICU (n=1742) | Age ≥65 years and in ICU (n=150) | |
| β-Lactams (antipseudomonal) | 178 (4.6)‡ | 21 (0. 5) | 407 (10.6)‡ | 58 (1.5) | 541 (29.0) |
| β-Lactams | 331 (8.6) | 9 (0.2) | 482 (12.5) | 20 (0.5) | 345 (18.5) |
| Fluoroquinolones | 502 (13.0) | 10 (0.3) | 273 (7.1) | 6 (0.2) | 252 (13.5) |
| Macrolides | 20 (0.5) | 0 (0.0) | 17 (0.4) | 0 (0.0) | 10 (0.5) |
| β-Lactams (antipseudomonal)+fluoroquinolones | 201 (5.2)‡ | 13 (0.3)‡ | 189 (4.9)‡ | 30 (0.8) | 238 (12.8) |
| β-Lactams+fluoroquinolones | 302 (7.8)‡ | 3 (0.1)‡ | 166 (4.3)‡ | 9 (0.2) | 177 (9.5) |
| β-Lactams+macrolides | 160 (4.2) | 2 (0.1) | 64 (1.7) | 2 (0. 1) | 55 (3.0) |
| β-Lactams (antipseudomonal)+macrolides | 50 (1.3)‡ | 0 (0.0) | 45 (1.2)‡ | 2 (0.1) | 58 (3.1) |
| Fluoroquinolones+macrolides | 24 (0.6) | 0 (0.0) | 11 (0.3) | 0 (0.0) | 6 (0.3) |
| Anti-MRSA drugs | 9 (0.2)‡ | 8 (0.2) | 12 (0.3)‡ | 6 (0.2) | 29 (1.6) |
| Others | 104 (2.7) | 13 (0.3) | 76 (2.0) | 17 (0.4) | 153 (8.2) |

*Data on empirical antimicrobial regimens in 112 patients were missing.

†Risk factors for pseudomonal infection was defined as chronic airway disease (bronchiectasis or chronic obstructive pulmonary disease) or healthcare-associated pneumonia according to Infectious Diseases Society America/American Thoracic Society criteria.[14]

‡Overtreatment was defined as (1) use of antipseudomonal β-lactams or β-lactams+fluoroquinolones in hospitalised (not in ICU) patients without risk factors for pseudomonal infection; (2) use of β-lactams (antipseudomonal or not)+fluoroquinolones in ICU patients aged <65 years without risk factors for pseudomonal infection; (3) use of anti-MRSA drugs in hospitalised (not in ICU) patients (use of anti-MRSA drugs in ICU patients with MRSA risk after influenza virus infection was considered adequate).[13]

CAP, community-acquired pneumonia; ICU, intensive care unit; MRSA.methicillin-resistant Staphylococcus aureus

discharge was 5 days. The median duration of ICU hospitalisation was 8 days. The top five causes of death were severe pneumonia/multiorgan dysfunction syndrome 69.1% (170/246), cardiac failure 2.8% (7/246), acute myocardial infarction 2.0% (5/246), stroke 1.6% (4/246) and gastrointestinal haemorrhage 1.6% (4/246).

Online supplementary appendix 8 shows the results of subgroup analysis of 30-day mortality. Fatality increased with age. Mortality was similar between male and female patients (4.9% vs 3.5%). Mortality in patients admitted to an ICU was 15.3%.

## DISCUSSION

This study represents the largest, multicentre, retrospective cohort study on the aetiologies and outcomes in adolescents and adults with CAP in China. In this study, we found that admission of patients with low mortality risk, inadequate microbiological diagnostic tests, overuse of antibiotics and incorrect serological testing for *M. pneumoniae*, *C. pneumoniae*, *L. pneumophila* and respiratory viruses were the main challenges of CAP management.

We identified four major categories of overuse of healthcare resources in CAP management in China:

1. A large number of low-risk patients were admitted to the hospitals. Guidelines for CAP management in China and the USA recommend that decisions for hospitalisation should be based on illness severity.[13 22]

It was estimated that over $8 billion are spent in CAP treatment every year in the USA, and the cost for inpatient CAP management is 25–30 times more than for outpatient CAP management.[23–25] Therefore, admission of low mortality risk patients with CAP results in major unnecessary cost expenditures. Moreover, outpatients usually return to their baseline activity levels much sooner than inpatients, and enjoyed a higher quality of life.[26 27] Finally, hospitalisation is associated with the risk of nosocomial infections, potentially caused by high-virulent and multidrug-resistant organisms.[28] Admission of low-risk patients with CAP was also observed in a recent large US study,[10] so it may not be unique to China. However, there are many other factors that play an important role in deciding the need for hospitalisation such as comorbidities, lack of available family support, older age, mental illness, drug abuse and so on.[29 30]

2. Length of stay in hospital was unnecessarily long. CAP guidelines recommend that patients should be discharged as soon as they achieve clinical stability and have no other active medical problems. Keeping patients in hospital and observing them while receiving oral antibiotic therapy or waiting for normalisation of all clinical parameters are not indicated and are associated with increased costs and potentially with inhospital adverse events.[12 28 29] We observed that patients

**Table 5** Supportive treatment and clinical outcomes of patients with CAP

| Items | Cases (%) |
|---|---|
| ICU admission | 367 (6.3) |
| Mechanical ventilation | |
| Non-invasive ventilation | 286 (4.9) |
| Invasive ventilation in ICU | 123 (2.1) |
| Invasive ventilation not in ICU | 33 (0.6) |
| Vasopressor use | 197 (3.4) |
| CRRT | 16 (0.3) |
| ECMO | 3 (0.1) |
| Systemic glucocorticosteroids use after diagnosis of CAP | 1540 (26.4) |
| ICU patients who received systemic glucocorticoids | 154 (2.6) |
| Patients on invasive mechanical ventilation who received systemic glucocorticoids | 75 (1.3) |
| Patients on non-invasive mechanical ventilation who received systemic glucocorticoids | 158 (2.7) |
| 30-day mortality | 246 (4.2) |
| Length of stay in hospital (days, median, IQR) | 11.0 (5.0–24.0) |
| Days between admission and clinical stability (median, n=5130, IQR) | 4.0 (1.0–10.0) |
| Days between clinical stability and discharge (median, n=5130, IQR) | 5.0 (1.0–9.0) |
| Length of stay in ICU (days, median, n=350, IQR) | 8.0 (4.0–16.0) |
| Treatment failure within 14 days | 427 (7.3) |
| Needs non-invasive ventilation | 169 (2.9) |
| Needs invasive ventilation | 145 (2.5) |
| Needs vasopressors | 130 (2.2) |
| Death | 147 (2.5) |
| Direct causes of death | |
| Severe pneumonia/MODS | 170 (69.1) |
| Heart failure | 7 (2.8) |
| Acute myocardial infarction | 5 (2.0) |
| Stroke | 4 (1.6) |
| Haemorrhage of digestive tract | 4 (1.6) |
| Acute renal failure | 2 (0.8) |
| Arrhythmia | 2 (0.8) |
| Accident aspiration | 1 (0.4) |
| Others | 51 (20.7) |

CAP, community-acquired pneumonia; CRRT, continuous renal replacement therapy; ECMO, extracorporeal membrane oxygenation; ICU, intensive care unit; MODS, multiple organ dysfunction syndrome

with CAP were discharged at a median of 5 days after achieving clinical stability, and 22% met clinical stability criteria at admission. Given the median LOS of 11 days for all patients with CAP, discharging patients with CAP once they achieved clinical stability would lead to cost savings of approximately half of the total hospitalisation expenses. Similarly, the length of stay in hospital may be influenced by other social factors.

3. 40.9% of patients without risk factors for pseudomonal infection received overtreatment with empiric antimicrobial regimens. Antipseudomonal β-lactams (28.2%) or β-lactams+quinolones (12.2%) were the most common empiric regimens for overtreatment. This may be due to overestimation of illness severity, clinicians unfamiliarity with CAP guidelines or lack of microbiological diagnostic testing. Moreover, we found quinolones use in more than 40% of patients with CAP. The US Food and Drug Administration has released warnings of potential adverse effects of fluoroquinolones, such as Q-T prolongation, tendon injury, psychiatric disorder and so on.[31–33] As second-line antituberculosis drugs, fluoroquinolones can also affect the diagnosis of tuberculosis and induce drug resistance.[34 35]

4. Incorrect serological testing was performed. We observed that many patients had an acute serum specimen collected for IgG serology testing for atypical bacteria and respiratory viruses without a convalescent serum specimen obtained for paired serological testing. Furthermore, many patients had testing for IgM antibodies for a variety of respiratory pathogens, but elevation of IgM antibodies with a low-normal IgG titre is uncommon during acute illness.[36–38] Paired serology for virus and atypical pathogens is recommended for epidemiological purpose. A follow-up convalescent serum specimen to document changes in IgG and IgM antibody levels is generally required for diagnosis.[39 40] Thus, the value of antibody testing on a single acute serum specimen to determine the aetiology of CAP is questionable. The costs of more frequent use of PCR testing on lower respiratory specimens may be partially offset by not performing serological testing in patients with CAP.

The strengths of this study, in contrast to some past epidemiological investigations,[41] included data on bacterial isolates obtained in current clinical practice, microbiological testing ordered and antimicrobials administered, according to Chinese standards of care, and the study population included adolescents and adults of all ages admitted to general hospital wards or ICUs from the participating centres to reduce selection bias. We also included patients who were critically ill, aged >90 years and with risk factors for HCAP.

This study had several limitations. First, given the retrospective study design, it is possible that selection bias was present and the study population may not have been representative of all patients with CAP admitted to the 13 participating sites. Second, the participating hospital sites were teaching hospitals in seven cities in three provinces, and were not selected to be representative of CAP hospital management in China, especially in smaller, rural hospitals. Third, this study reports on CAP management during 2014; analysis of multiple years of data can allow assessment of changes in CAP management. Fourth, 45.7% of patients with CAP received antibiotics before hospital admission and specimen collection, which may reduce the detection of some bacterial infections, such as *S. pneumoniae*. The low number of tests performed (good-quality sputum, blood cultures, urine antigens, PCR) limit the knowledge of the true aetiology of CAP

in the study. Therefore, the bacterial pathogens identified in this study may not be representative of all bacterial causes of CAP in the source patient populations for this study. Finally, while we included adolescents, the majority of patients were adult patients with CAP, and our findings do not apply to children hospitalised with CAP.

In conclusion, we characterised adolescents and adults hospitalised for CAP in China and identified several problems suggesting the overuse of healthcare resources in CAP management. This suggests that education and training of clinicians on current CAP guidelines in China are needed to improve clinical management and could also result in substantial cost saving in healthcare expenditures for patients with CAP. The multicentre hospital network can serve as a platform for conducting intervention studies for hospitalised patients with CAP in the future, using the baseline data from this observational study.

**Author affiliations**
[1]Department of Infectious Diseases and Clinical Microbiology, Beijing Chao-Yang Hospital, Capital Medical University, Beijing, China
[2]Department of Infectious Disease, 4th Medical College of Peking University, Beijing Jishuitan Hospital, Beijing, China
[3]Department of Pulmonary and Critical Care Medicine, Center for Respiratory Diseases, National Clinical Research Center of Respiratory Diseases, China-Japan Friendship Hospital, Beijing, China
[4]Department of Respiratory Medicine, Yan'an Hospital, Kunming Medical University, Kunming, China
[5]Department of Respiratory Medicine, Qingdao Municipal Hospital, Qingdao, China
[6]Department of Respiratory Medicine, Beijing Huimin Hospital, Beijing, China
[7]Department of Respiratory Medicine, Linzi District People's Hospital, Zibo, China
[8]Department of Respiratory Medicine, Beijing Luhe Hospital, Capital Medical University, Weifang, China
[9]Department of Pulmonary and Critical Care Medicine, Weifang NO.2 People's Hospital, Weifang, China
[10]Department of Respiratory Medicine, Qilu Hospital Of Shandong University (Qindao), Qingdao, China
[11]Department of Respiratory Medicine, The 2nd Hospital of Beijing Corps, Chinese Armed Police Forces, Beijing, China
[12]Department of Infectious Disease, Yantai Yuhuangding Hospital, Yantai, China
[13]Department of Respiratory Medicine, Rizhao Chinese Medical Hospital, Shandong Chinese Medical University, Rizhao, China
[14]Occupational Medicine and Toxicology Department, Beijing Chao-Yang Hospital, Capital Medical University, Beijing, China
[15]Influenza Division, National Center for Immunization and Respiratory Diseases, Centers for Disease Control and Prevention, Atlanta, Georgia, USA
[16]Departmentof Pulmonary Medicine, Capital Medical University, Beijing, China

**Acknowledgements** We thank Jay Purdy (senior director, Anti-infectives, Pfizer, 500 Arcola Rd, F3203, Collegeville, PA 19426, USA), Francesco Blasi (Respiratory Unit, IRCCS Fondazione Ca' Granda Ospedale Maggiore Policlinico, Department of Pathophysiology and Transplantation, University of Milan, Milan, Italy) and Richard G Wunderink (Division of Pulmonary and Critical Care Medicine, Department of Medicine, Northwestern University Feinberg School of Medicine, Chicago, Illinois, USA) for their valuable comments on the manuscript.

**Contributors** Study design: LC, FZ, HL, CW, BC. Data collection: LC, FZ, HL, XX, XH, YW, CZ, LS, JW, GY, GW, XY, HY, LW, ML, CX, BL, XZ, YL, YX, XC, LL. Statistical analysis: LC, FZ, HL, XH. Writing: LC, BC, TMU. All authors take full responsibility for the study design, data analysis and interpretation, and preparation of the manuscript. All authors approved the final draft manuscript.

**Funding** This work was funded by National Science Grant for Distinguished Young Scholars (81425001/H0104), National Key Technology Support Program from Ministry of Science and Technology (2015BAI12B11) and The Beijing Science and Technology Project (D151100002115004).

**Disclaimer** The views expressed are those of the authors and do not necessarily reflect the official policy of the Centers for Disease Control and Prevention.

**Competing interests** None declared.

**Patient consent** Not required.

**Ethics approval** The Ethics Committee of China-Japan Friendship Hospital (no. 2015-86)

**Provenance and peer review** Not commissioned; externally peer reviewed.

**Data sharing statement** No additional data available.

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
