## [Reviewer comments · BMJ Open]

ARTICLE DETAILS

TITLE (PROVISIONAL)	Disease characteristics and management of hospitalized adolescents and adults with Community-Acquired Pneumonia in China : a retrospective multicenter survey
AUTHORS	Chen, Liang; Zhou, Fei; Li, Hui; Xing, Xiqian; Han, Xiudi; Wang, Yiming; Zhang, Chunxiao; Suo, Lijun; Wang, Jingxiang; Yu, Guohua; Wang, Guangqiang; Yao, xuexin; Yu, Hongxia; Wang, Lei; Liu, Meng; Xue, Chunxue; Liu, Bo; Zhu, Xiaoli; Li, Yanli; Xiao, Ying; Cui, Xiaojing; Li, Lijuan; Uyeki, Timothy; Wang, Chen; Cao, Bin

VERSION 1 – REVIEW

REVIEWER	Dr MA Hon Ming Hong Kong, SAR, China
REVIEW RETURNED	01-Aug-2017

GENERAL COMMENTS	1. In the Method part, why the age of 14 years was chosen as the lower cut-off for case recruitment?2. In the Method part, was nasopharyngeal (NP) swab or aspirate be performed routinely for CAP or HCAP patients? In the Result part, the results of NP swab or aspirate are not mentioned.3. In the Result part, what percentage of patients had the order of do-not-resuscitate (DNR)?4. In Table 1, what were the risk factors for aspiration? Please add a footnote in the table.5. In Discussion part, over-hospitalisation based on the severity of pneumonia or other risk factor is an over-generalised statement. There are many other factors that play an important role in deciding the need for hospitalisation, such as comorbidities and availability of caregiver. I would say they are potential avoidable. The same principles are applied to the length of hospital stay.6. In the Discussion part, the paired serology for virus and atypical pathogens is recommended for epidemiological purpose. It is not strictly indicated for treatment. Many a times, the convalescence titre is not checked because the patients have been discharged already and no follow-up is arranged for this purpose. We can't say the acute titre is unnecessary.
--

REVIEWER	Diego Viasus Universidad del Norte, Colombia
REVIEW RETURNED	11-Aug-2017

GENERAL COMMENTS	I am very grateful for the invitation to review this manuscript. The authors conducted a retrospective study of adult patients requiring hospital admission with community-acquired pneumonia in China. Comments  1. The inclusion of immunosuppressed patients makes the interpretation of the data difficult. In the majority of studies of community-acquired pneumonia (as well as guideline recommendations) are based on immunocompetent patients. Therefore, the authors should perform an analysis of the immunocompetent patients, to make the data comparable with studies from other regions of the world. 2. How can the authors explain the high frequency of Gram-negative causative pathogen. This result does not correspond to data found in the majority of studies of community-acquired pneumonia. 3. How was defined "illness onset in the community"? 4. Why was fever defined as "temperature ≥ 37.3 ° C"? 5. Why authors included "HCAP according to IDSA / ATS criteria" as a risk factor for Pseudomonas spp.? 6. Some patients with low CURB65 are hospitalized for reasons other than the value of the scale (eg low oxygen saturation or multilobar pneumonia). The authors should clarify this point due that some of this information could be available. 7. Based on the discussion section, the term "unnecessary" in relation to the use of serological tests should be modified to "inadequate" or "incorrect".
---

Reviewer: 1

Reviewer Name: Dr MA Hon Ming

Institution and Country: Hong Kong, SAR, China

Please state any competing interests or state 'None declared': None declared

Please leave your comments for the authors below

1. In the Method part, why the age of 14 years was chosen as the lower cut-off for case recruitment?

Authors' response: Thanks for your comments. In mainland of China, patients <14yrs were treated by pediatricians, while patients aged >14yrs were classified as adolescents/adults and asked to see general clinicians. This is the reason why 14 year was chosen as the cutoff in our study.

2. In the Method part, was nasopharyngeal (NP) swab or aspirate be performed routinely for CAP or HCAP patients? In the Result part, the results of NP swab or aspirate are not mentioned.

Authors' response: Thank you comments. In Method part, we have made it clear that nasopharyngeal (NP) swab was used for antigen testing for Influenza A and Influenza B. Aspirate was not routinely used for antigen testing. In the Result Part, results of NP swab have been added. About 0.7% and 0.4% patients had nasopharyngeal swab antigen testing for Influenza A and Influenza B respectively.

3. In the Result part, what percentage of patients had the order of do-not-resuscitate (DNR)?

Authors' response: Thanks for your question. In this survey, we did not have the data of order of do-not-resuscitate (DNR).

4. In Table 1, what were the risk factors for aspiration? Please add a footnote in the table.

Authors' response: In our study, the risk factors for aspiration included choking, drowning, nasal feeding, pseudobulbar palsy, dementia, coma, poisoning, Parkinson's disease. We have added this information as footnote in Table 1.

5. In Discussion part, over-hospitalisation based on the severity of pneumonia or other risk factor is an over-generalised statement. There are many other factors that play an important role in deciding the need for hospitalisation, such as comorbidities and availability of caregiver. I would say they are potential avoidable. The same principles are applied to the length of hospital stay.

Authors' response: Thanks for your kind suggestions. We have modified the discussion part according to your comments by adding other factors, such as comorbidities, lack of available family support, older age, mental illness and drug abuse, together with severity of pneumonia for decision of hospitalization and length of stay.

6. In the Discussion part, the paired serology for virus and atypical pathogens is recommended for epidemiological purpose. It is not strictly indicated for treatment. Many a times, the convalescence titre is not checked because the patients have been discharged already and no follow-up is arranged for this purpose. We can't say the acute titre is unnecessary.

Authors' response: Thanks for your comments. We totally agree with you that paired serology for virus and atypical pathogens is recommended for epidemiological purpose. Use of acute titre for guiding treatment is incorrect. We have modified the Discussion part accordingly.

Reviewer: 2

Reviewer Name: Diego Viasus

Institution and Country: Universidad del Norte, Colombia

Please state any competing interests or state 'None declared': None declared

Please leave your comments for the authors below

I am very grateful for the invitation to review this manuscript. The authors conducted a retrospective study of adult patients requiring hospital admission with community-acquired pneumonia in China.

Comments

1. The inclusion of immunosuppressed patients makes the interpretation of the data difficult. In the majority of studies of community-acquired pneumonia (as well as guideline recommendations) are based on immunocompetent patients. Therefore, the authors should perform an analysis of the immunocompetent patients, to make the data comparable with studies from other regions of the world.

Authors' response: Thank you for your comments. In this revised paper, we have excluded immunocompromised population (266 cases) and recalculated all the values. Because these patients account for only 4.4% of the whole population, the final conclusion is not changed.

2. How can the authors explain the high frequency of Gram-negative causative pathogen. This result does not correspond to data found in the majority of studies of community-acquired pneumonia.

Authors' response: Thanks for your comments. 45.7% of CAP patients received antibiotics before hospital admission and before specimen collection, which may reduce the detection of some bacterial infections, such as *Streptococcus pneumoniae*. Urinary antigen testing for *Streptococcus pneumoniae* was performed only in 2.6% of total population. Therefore, the bacterial pathogens identified in this study may not be representative of all bacterial causes of CAP in the source patient populations for this study.

To make the statement more accurate, we have added the limitation in Discussion part.

3. How was defined "illness onset in the community"?

Authors' response: defined as community acquired infection among those who have not been hospitalized during recent 28 days [11].

[11] Jain S, Self WH, Wunderink RG, et al. Community-Acquired Pneumonia Requiring Hospitalization among U.S. Adults. *N Engl J Med*. 2015;373:415-27.

4. Why was fever defined as "temperature ≥ 37.3 ° C"?

Authors' response: Thanks for your question. There is no agreement of normal temperature range among different population in indifferent studies. Fever definition of axillary temperature ≥ 37.3 ° C is well accepted and used in clinical practice among Chinese population. In a study on hospitalized low-risk community-acquired pneumonia patients in China, the authors also used the fever definition of axillary temperature ≥ 37.3 ° C [12].

[12] Zhou QT, He B, Zhu H. Potential for cost-savings in the care of hospitalized low-risk community-acquired pneumonia patients in China. *Value Health*. 2009;12:40-6.

5. Why authors included "HCAP according to IDSA / ATS criteria" as a risk factor for *Pseudomonas* spp.?

Authors' response: Thanks for your comments. Although controversy exists on the prediction value of HCAP for multidrug-resistant infection, there is consistent evidence that a percentage of patients with

community-onset pneumonia fulfilling the HCAP criteria have multidrug-resistant infection [16-18]. In a retrospective study among hospitalized patients with HCAP over 10 years in China, the most common microorganisms detected were *P. aeruginosa* [17]. As the spectrum of etiology and drug-resistance issue of CAP is regional, we think HCAP is one of risk factors for *Pseudomonas* spp.

[16]. Maruyama T1, Fujisawa T, Okuno M, et al. A new strategy for healthcare-associated pneumonia: a 2-year prospective multicenter cohort study using risk factors for multidrug-resistant pathogens to select initial empiric therapy. *Clin Infect Dis*. 2013;57:1373-83.

[17]. Qi F, Zhang GX, She DY et al. Healthcare-associated Pneumonia: Clinical Features and Retrospective Analysis Over 10 Years. *Chin Med J (Engl)*. 2015;128:2707-13.

[18]. Seong GM, Kim M, Lee J, et al. Healthcare-Associated Pneumonia among Hospitalized Patients: Is It Different from Community Acquired Pneumonia? *Tuberc Respir Dis (Seoul)*. 2014;76:66-74.

6. Some patients with low CURB65 are hospitalized for reasons other than the value of the scale (eg low oxygen saturation or multilobar pneumonia). The authors should clarify this point due that some of this information could be available.

Authors' response: Thanks for the comments. We agree with you that there are many factors play an important role in decision of hospitalization and length of stay in hospital, such as comorbidities, availability of caregiver and family support, et al. We have modified in the Discussion part.

7. Based on the discussion section, the term "unnecessary" in relation to the use of serological tests should be modified to "inadequate" or "incorrect".

Authors' response: Thanks for your suggestions. In the revised paper, we have replaced "unnecessary" by "incorrect"

VERSION 2 – REVIEW

REVIEWER	Dr Ma Hon-Ming The Chinese University of Hong Kong Prince of Wales Hospital HKSAR, China No Competing Interest
REVIEW RETURNED	27-Sep-2017
GENERAL COMMENTS	N/A

REVIEWER	Diego Viasus
-----------------	--------------

	Universidad del Norte and Hospital Universidad del Norte, Colombia I declare that I have no conflict of interest
REVIEW RETURNED	24-Sep-2017

GENERAL COMMENTS	 1. Due to an inadequate use of diagnostic tests documented in the study, this topic should be added in the abstract. 2. page 6, line 54: please write the diseases that were considered as immunosuppression. 3. Add data about the use of anti-MRSA drugs in hospitalized patients. Was it adequate? 4. Page 23, line 27: microbiological tests were not a strength of this study. The low number of tests performed (good quality sputum, blood cultures, urine antigens, polymerase chain reaction) limit the knowledge of the true etiology of CAP in the study. The above must be added to the limitations of the study.
--

VERSION 2 – AUTHOR RESPONSE

Reviewer 2

1. Due to an inadequate use of diagnostic tests documented in the study, this topic should be added in the abstract.

Authors' response: Thanks for your comments. Inadequate diagnostic tests were documented in the study, and a definite or probable pathogen was identified only in 12.7% of patients (738/5828). We have added this in the abstract.

2. page 6, line 54: please write the diseases that were considered as immunosuppression.

Authors' response: Thanks for your comments. We have added the diseases that were considered as immunosuppression, including HIV(+), chemotherapy/radiotherapy within 6 months, immunosuppressive therapy, organ/bone marrow transplantation, splenectomy, hematological neoplasms.

3. Add data about the use of anti-MRSA drugs in hospitalized patients. Was it adequate?

Authors' response: Thanks for your comments. According to China CAP guideline, use of anti-MRSA in ICU patients with MRSA risk after influenza virus infection is considered adequate. But use of anti-MRSA drugs in hospitalized (not in ICU) patients was regarded as overtreatment.

4. Page 23, line 27: microbiological tests were not a strength of this study. The low number of tests performed (good quality sputum, blood cultures, urine antigens, polymerase chain reaction) limit the knowledge of the true etiology of CAP in the study. The above must be added to the limitations of the study.

Authors' response: Thanks for your comments. We have added the low number of tests performed as the 4th limitation (Page 23, line 20-22).